# Trade Policy Uncertainty and Medical Innovation: Evidence from Developing Nations

Muhammad Nadir Shabbir [1],[*] , Wang Liyong [1] and Muhammad Usman Arshad [2]

1 School of International Trade and Economics, Central University of Finance and Economics, Beijing 100081, China
2 Department of Commerce, University of Gujrat, Hafiz-Hayat Campus, Gujrat 50700, Pakistan
* Correspondence: muhammadnadir948@gmail.com or 2019630007@email.cufe.edu.cn

**Abstract:** This study explores the influence of trade policy uncertainty on medical innovation investment in developing nations from 1980 to 2020, with a focus on the period of COVID-19. We used exogenous and heterogeneous exposure to trade-policy-uncertainty resolutions from developing countries' trade policy adjustments, which reduced tariff hikes on imported goods in a double difference-in-differences method. ARDL with PVAR has been studied for long-run and short-run analyses. The findings revealed that reducing tariff uncertainty boosts innovation beyond patent filings and margin reaction and exports. Long-term impacts of sectoral innovation patterns, governmental changes, and foreign technology entering developing nations have little effect on the findings. This paper also shows a long-term link between medical innovation, trade policy uncertainty, and research-and-development spending. Innovation's negative response to the innovation shock and research and development's positive response corroborates bidirectional and unidirectional causality. This study contributes to medical innovation and policy uncertainty in terms of developing countries and, most importantly, in trends of medical innovation, contemporaneous policy uncertainty given the inflow of foreign technology, and the importance of that technology recent times.

**Keywords:** trade policy; uncertainty; innovation; patents; developing nations

**JEL Classification:** D72; F13; F14; O19; O24; O33; P33





## 1. Introduction

This study examines how trade policy uncertainty affects medical innovation investment in developing nations. Companies may benefit from delaying investment until business conditions improve, as has been described well by Bernanke (1983), Dixit (1989), Dixit et al. (1994), and Rodrik (1991). Emerging empirical literature explains that firms' investment behavior is consistent with this basic mechanism (Baker et al. 2016; Gulen and Ion 2016; Handley and Limão 2012, 2017; Julio and Yook 2016; Koijen et al. 2016). However, most empirical research focuses on employment, physical capital, productivity, or economic sectors, whereas investment in innovation or innovation across industries has received little attention.

Innovation, important to economic growth, is hampered by policy ambiguity, and policy uncertainty has accelerated in recent years. These two reasons make it vital to evaluate policy uncertainty's impact on innovation (Baker et al. 2016). The US–China trade conflict, Brexit, the North American Free Trade Agreement (NAFTA) renegotiation, and medical innovation pertaining to COVID-19 have all led to the increase of tariffs being a commercial concern. Investors, firms, and monetary authorities face the biggest global recession risk due to the trade war roller coaster. Some say tariff uncertainty is worse than actual tariffs. Handley and Limão (2015) adhere to the primary theoretical framework they created in 2014. In exchange for less protection, preferential trade agreements (PTAs)

abolish some protections. There were 283 PTAs in July 2010, a significant rise from 1990. This demonstrated that trade policy uncertainty decreases investment and entry into export markets and that exporters can benefit from PTAs even when trade barriers are low or nonexistent.

Coelli et al. (2016) used international firm-level patent data to quantify the impact of 1990s-era trade policy on innovation in 60 countries as trade liberalization influences innovation, affecting technical change and growth. Trade policy liberalization likely boosted knowledge generation by 7% in the 1990s; increased patenting demonstrated innovation, not just information protection. It concluded that trade liberalization improves market access and import competition, boosting innovation. Amiti et al. (2017) used Chinese firm-product data for 2000–2006 to investigate the country-to-country impact of trade policy on innovation. This aggregate model predicts exporter pricing and quantity due to World Trade Organization (WTO) participation. The authors say that China's reduced input tariffs are the key source of U.S. welfare gains from China's entry into the WTO.

Coelli (2018) explored the exogenous and heterogeneous sensitivity to trade-policy-uncertainty resolutions after the U.S. discontinued tariff increases on Chinese imports. They documented that reducing tariff uncertainty has an economically and statistically significant effect. Moreover, the results are robust even when sectoral innovation trends, simultaneous policy changes, and foreign technology flow into China are considered. Health benefits and industrial growth depend on medical innovation and demand. Information, procedures, medications, biologics, technologies, and services help prevent and treat old and emerging diseases. This boom in medical innovation explains healthcare's rapid growth, as asserted by some authors (Cutler 1995; Fuchs 1996; Newhouse 1992). Understanding the rise of this enterprise and the medical research and development expenditures (R&D) that propelled it requires understanding investor returns.

According to Newhouse (1992), most countries invest in new technologies, medical services, and hospitals, as well as in medical innovation, which health economists argue is the driving force behind the rise in global health care spending. Innovation is stimulated by direct and indirect payments (prescription medicine reimbursement and indirect reimbursements are for medical devices). According to Becker et al. (2005) and Murphy and Topel (2006), advances in healthcare have led to economic growth and decreased worldwide inequality. By using data from developing nations, it can be seen that better healthcare is comparable to other forms of economic expansion over the past century, as measured by per capita income (GDP). Increased life expectancy and quality may be the most valuable shift of the century; thus, the size and growth of the healthcare sector have spurred public debate. Medical equipment, biologics, medications, and associated services drive U.S. healthcare expenses. Private and public reimbursement restrictions affect U.S. profitability, and the Centers for Medicare Services (CMS) says Medicare and Medicaid funded 44% of U.S. spending in 2012, and Europe's government pays 85% of healthcare costs.

Because health care manufacturing is primarily unfunded by public markets, manufacturers use public capital markets to fund R&D. Hospitals pay 35% of the total cost of health care and rely on debt or donations to operate. Twenty-two percent of healthcare expenditures are attributable to privately funded clinics; consequently, for-profit medical innovation companies are overrepresented on public stock markets due to a lack of public equity financing in important healthcare fields. Due to the concentration of U.S. medical product sales, U.S. government policies influence medical R&D returns. Egan and Philipson (2013) estimate that in 2012, U.S. health care spending accounted for 48% of worldwide spending, although U.S. GDP accounted for only 24% of global GDP. The United States' proportion of worldwide spending on biopharmaceuticals is 39%, as many rising nations spend more. Due to greater markups, U.S. markets contribute more to overall earnings than sales, concluding that medical R&D requires payment modifications that risk U.S. markups, whereas U.S. reimbursement policies impact asset values.

It is widely acknowledged that R&D expenditures in medical innovation are driven by global returns rather than returns on the domestic market. For example, Swedish medical

product companies innovate to sell globally, not just domestically, because global returns stimulate innovation and boost healthcare spending. A country's healthcare economy and policies affect its growth. A tiny European country's growth depends on how U.S. policies affect global returns, as do future Medicare expenditures. Few health economists have studied how one country's healthcare policies affect another. Hammar and Belarbi (2021) study the non-linear relationship between R&D spending and innovations like productivity and high-tech exports. It has been demonstrated that linearity is frequently conditioned by other macroeconomic factors like the degree of development and financial openness. According to the findings, the R&D, innovation, and productivity threshold effects are strongest in the U.S., and the data shows that R&D spending, innovation, productivity, and medium- and high-tech exports have mixed effects. In contrast, positive and negative effects depend on innovation indicators or threshold variable levels. Therefore, the findings support the notion that the level of economic development can be used as a target indicator for implementing an innovation policy.

This work utilizes medical patent applications from 65 developing nations from 1980 to 2020. Almost every company submits a patent that is tracked, including the filing date and the patent's technical class, which we link to product codes, and the filing country. We generated a panel dataset on the patenting of medical technologies by using these data. The empirical strategy reduces industry-specific innovation by utilizing the differential between "column 2" and most favored nation (MFN) tariffs. We compare innovation in the uncertain medical industry before and after permanent-normal trade relations (PNTR) (the second difference) to analyze R&D and medical innovation, whereas innovation strategy takes into account industry and tech developments, which is a significant contribution to this study. Nonetheless, patentability and sunk R&D vary by industry, product, and time; industry-fixed effects erase only time-variable variations. Finally, to capture dynamic interdependencies and reverse causality, the panel vector autoregressive (PVAR) model will be implemented in the investigation of the timeliness of technological innovation.

The remainder of this study follows this pattern. Section 2 will discuss the literature review and hypothesis development, which will help the author design a fundamental analysis. Section 3 discusses the economic framework of study, and Section 4 discusses the research methodology and design, including the population and sample size, variable and model descriptions, and data analysis technique. The results and discussion are discussed in Section 5, and the conclusion and policy implications and limitations of study are discussed in Section 6.

## 1.1. Real-World Case Study of Australia

Australia has high-quality data and a policy variation relevant to uncertainty (Handley and Limão 2015). Australia has historically had significant trade barriers. Unilateral liberalization has created enormous gaps between protection and obligations. It recently imposed preferential trade agreements (PTAs) like many developed countries. Several agreements were with developing countries with preferential, discretionary market access. These considerations include various trade-policy-uncertainty sources from the start. Theoretically informed empirical assessments of Australia's policy uncertainty and trade policy tools should be valid in a variety of applications and policy negotiations.

Australia currently has low tariffs, but this has not always been the case from a historical perspective. Lloyd's (2008) study of 100-year time series for Australian tariffs demonstrates that some sectors were heavily protected in the early 1990s. Pre- and post-war protectionism and political meddling in tariff-making left a legacy according to Glezer (1982). The late 1980s and 1990s saw gradual, unilateral liberalization. Even in low-tariff sectors, a 2002–2006 exporter could look back a decade and fear a high-tariff system.

Australia's binding commitments are large and dispersed due to the Uruguay Round (1986–1994) of multilateral negotiations (Corden 1996). Many products have zero or near-zero imposed tariffs, whereas maximum bound rates range from 0% to 55%. This variance in the applied-to-bound gap is empirically utilized. In a procedure called "tariffication,"

Australia lifted most import quotas and restrictions after the Uruguay Round negotiations (Snape et al. 1998). Trade barriers are now measured uniformly across products.

Australia's Productivity Commission cited backsliding n preferential trade agreements and liberalization. The Commission's review of Australia's trade agreements notes that even if agreements don't reduce existing barriers, they can lock in present policies, preventing countries from adding hurdles in the future Productivity Commission (2010).

If Australia reverted all tariffs to their original form, the tariff profile would change. Only 24% of Australia's MFN tariffs match the contractual pledge in 2004. Reversing bindings can cause huge changes. Figure 1's histogram reveals that 73% of MFN tariffs could rise, some by 35%. Such reversals might diminish an exporter's profits by 19% annually. Figure 1 illustrates that profit losses are widespread. A full reversal to binds would reduce profit distribution from 2004 levels.

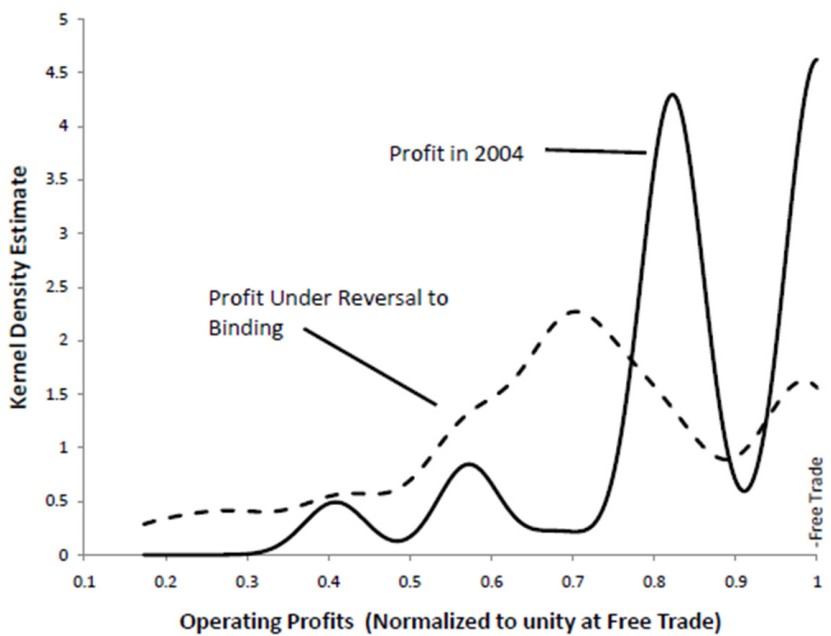

**Figure 1.** Shift in distribution of profits under a binding reversal in 2004 at applied (MFN rate) vs. bound tariff levels.

### 1.2. COVID-19 and Uncertainty

In the present time, COVID 19 is the factor that affects the medical industry and medical innovation. Innovation in the medical sector accelerated in the pandemic era and has had a huge impact on trade uncertainty, innovation, and investment, which is very important in the medical sector. After post-PNTR time, pre- and post-COVID time is also a recognizable shock to medical sector innovation.

The COVID-19 pandemic raised uncertainty in many elements of daily life (Caggiano et al. 2020), and many characteristics of the virus remain unknown (Fauci et al. 2020). No one knows when the world will return to normal; the authors underline the significance of global cooperation and the importance of international governmental, corporate, and non-profit sectors working together to continue manufacturing vaccines (Corey et al. 2020; Gates 2020). Lockdowns and quarantines exacerbated stress and fear in many countries (Qiu et al. 2020). Scarce medical supplies, such masks and ventilators, have prompted nations to compete for them, forcing hospitals and health institutions to ration their inventory. These reasons have stoked global unease.

Baker et al. (2020a) show that present uncertainty levels are higher than during the 2008–2009 Great Recession and are closer to the Great Depression. They also say the current economic slowdown is due to COVID-19's excessive uncertainty. Sharif et al. (2020) validate

COVID-19's impact on political and regulatory uncertainties. Albulescu (2020) notes that everyday statements about infections and deaths boost EPU.

High uncertainty can complicate enterprises' activities by causing them to delay investment decisions (Chu and Fang 2020) and assume less debt (Guo et al. 2020), which could worsen the economic crisis and reduce cash injections. Baker et al. (2020b) said no disease has ever affected the stock market as much as COVID-19.

These findings support the theory that COVID-19 uncertainty produced poorer economic growth, above-average bankruptcy rates, and high unemployment. This pandemic's unpredictability has deterred government authorities, corporate executives, and even individuals from making decisions. This complicates decision-making for private, governmental, and nonprofit executives.

## 2. Literature Review and Hypothesis Development

This study examines the relationship between trade policy uncertainty, spending on R&D, and innovation in the medical sector of industrialized and developing nations. It describes the economic process and the rationale for the empirical study. Incorporating uncertain technology options into a commerce model with heterogeneous firms produces a significant discovery. We build upon Handley and Limão's (2018b) work and focus on the innovation decision of the firm. The entire economic system is dominated by monopolies. Companies can increase their productivity by compensating the government for sunk investment costs. In Bustos (2011), for instance, binary technology is selected.

### 2.1. Trade Policy Uncertainty, Research and Development, and Innovation

The literature on new product and service development stresses market size. Even before the PNTR, the United States was China's most important market, but the possibility of surprise tariff increases may have prompted enterprises to defer exporting and development. Companies began exporting after adopting permanent MFN status, according to Handley and Limão (2018a). Exporting companies should patent their products in the United States. This is determined by comparing the number of U.S. patent applications filed in highly exposed industries. According to our findings, more patent applications filed and granted by the USPTO result in a higher PNTR (USPTO). Due to the high expense of patenting and the fact that companies tend to file patents abroad only if they intend to export, increased U.S. exports are stimulating innovation (Coelli et al. 2016).

Two scholarly research strands inform the study. First, this study was informed by the real options in the literature that state that uncertainty generates an option value of waiting, delaying irreversible investment (partially). Recent studies of uncertainty's effect on investment behavior include those of Bernanke (1983); Bloom (2007, 2016); Bloom et al. (2018); Dixit (1989); Dixit et al. (1994); Guiso and Parigi (1999); Rodrik (1991). Other research has analyzed and empirically examined the effects of policy uncertainty on investment (Baker et al. 2016; Fernandez-Villaverde et al. 2011; Gulen and Ion 2016; Handley and Limão 2012, 2018a; Julio and Yook 2016; Koijen et al. 2016). Most research on this topic has centered on physical capital investment, employment, and productivity, whereas the effects of uncertainty on R&D and innovation have been largely neglected. Exceptions exist, as Bloom (2007) demonstrates that R&D is less sensitive to shifts in demand under high levels of uncertainty, but he offers no actual evidence. Koijen et al. (2016) examine the U.S. healthcare business and demonstrate that government-induced uncertainty boosts medical innovation while decreasing medical R&D. After receiving the PNTR funding, Handley and Limão (2018b) show indirect evidence of technical changes that reduced marginal export costs. This research fills a gap in the literature by demonstrating that removing trade policy uncertainty increases innovation investment across industries.

In addition, the contemporary literature on heterogeneous firms and trade emphasizes the complementarity between increasing overseas market access and investment in productivity-enhancing activities (Atkeson and Burstein 2010; Costantini and Melitz 2009; Lileeva and Trefler 2010). Other studies consider the impact of exporting on productivity

(Bernard and Jensen 1999; Biesebroeck 2005; Clerides et al. 1998; De Loecker 2007; Yan et al. 2022). Instead of discussing export status and productivity, this article focuses on investment in trade policy and innovation. In previous literature, uncertainty represented the most substantial shift. Based on Handley and Limão (2018a) and Bustos (2011), we present technology selection under uncertainty in a dynamic commerce model with heterogeneous firms. The approach combines two legitimate alternatives: option value of waiting and market access (Lileeva and Trefler 2010). The model, similar to that of Bustos (2011), argues that the decision to innovate is driven endogenously by market size and that only the most productive enterprises benefit from innovation. In contrast to Bustos (2011), the cost of innovation has already been invested, and uncertainty regarding foreign trade policy causes a "band of inaction" in which businesses do not invest and maintain a low level of technology. Reducing uncertainty decreases the option value of waiting, encouraging more businesses to innovate, and facilitating the development of the following hypothesis.

**H1.** *Trade policy uncertainty (TPU) and research and development (R&D) significantly affect patents (innovation) under the DID perspective.*

Shea (1999) describes how patents or R&D spending might be used for long-term innovation analysis. She discovered a modest link between total factor productivity (TFP) and technology shocks after developing direct measurements of technological innovation based on patents and R&D spending. The results of Shea (1999) will likely demonstrate the limitations of basic patent counts, as they do not account for the vast economic variability of patents (Griliches 1998; Kortum and Lerner 1998). It is assumed that landscape changes, whether slow changes, as defined by Geels (2002), or shocks, as discussed by van Driel and Schot (2005), exert pressure on incumbent regimes, creating openings for radical niche innovation. On the other hand, it is generally recognized that not all economic crises or global shocks result in innovation breakthroughs, especially in the short term. Rather, such crises may have undesirable outcomes (Geels 2002).

Identifying breakthrough inventions as the most important patents, i.e., patents in the right tail of our measure, enables us to develop aggregate and sectorial indices of technical change (Kelly et al. 2021). Their technological indexes range from 1840 to 2010 and include innovations from private and public companies, non-profit organizations, and the United States government. These indices properly depict the evolution of technological waves across time and are outstanding predictors of future output. The objective of the research conducted by Taalbi (2021) was to pit historical narratives and periodization of technological transitions against systematic data based on patterns in innovation output and innovation biographies. This research used data on innovation output and examined two concerns concerning the long-term innovation trends in the industrial industry, and paved the way to develop the following hypothesis.

**H2.** *Trade policy uncertainty (TPU) and research and development (R&D) significantly affect patents (innovation) under the short- and long-run perspective.*

### 2.2. Threshold Effect and VAR Model

Current studies on trade policy uncertainty fall into the following categories: First, they quantify the degree of trade policy uncertainty. Baker et al. (2016) derived the economic policy uncertainty index by using news articles and discovered that it is highly consistent with macrofluctuations, which explains its rationale. Caldara et al. (2020) utilised quarterly earnings call transcripts to identify trade policy uncertainty at the firm level. Tariff measurement assumes tariffs are the only source of trade policy uncertainty and analyses commerce before and after policy changes.

Secondly, the current studies focus on the macroeconomic implications of trade policy insecurity, imports and exports. By using Australian data, Handley et al. (2013) examined the effect of trade policy uncertainty on export enterprises. Product variety would have decreased 7% from 1993 to 2001 without WTO-imposed binding obligations. Removed uncertainty would account for more than half of Australia's predicted new product growth if all tariffs and restrictions are eliminated. Chen and Zhao (2021) and Mao (2020) used the DID technique to analyze the impact of trade policy uncertainty on Chinese companies' imported goods as a quasi-natural experiment. Reducing trade policy uncertainty has been shown to enhance imports in terms of volume, likelihood, duration, and quality.

China's joining the World Trade Organization (WTO) in 2001 reduced trade policy uncertainty. It increased the number of trade items available to Chinese businesses, which increased firm employment. In turn, this increased the number of employees in Chinese businesses. A study by Pierce and Schott (2016) found that China's reduced external trade policy uncertainty harmed U.S. manufacturing jobs after PNTR was given to China in 2001. Shepotylo and Stuckatz (2017) found that trade policy uncertainty impacted foreign investment as EU's investment in Ukraine would increase if trade policy uncertainty in Ukraine were reduced. Caldara et al. (2020) looked at the macro and local effects of a trade policy uncertainty shock. Firm-level estimates imply that trade policy uncertainty may have reduced aggregate U.S. investment by 1% in 2018, whereas aggregate evidence from VAR analysis shows a negative impact of 1.5% to 2% on private investment in the United States in 2018.

Thirdly, research is conducted on the microeconomic implications of trade policy uncertainty and procurement patterns. Heise et al. (2017) analyzed the effect of trade policy uncertainty on company procurement patterns. They discovered that when trade policy uncertainty is high, businesses choose American-style procurement to avoid trade conflicts. When trade policy uncertainty is low, the likelihood of a trade war is low, and corporations choose Japanese-style procurement because it is less expensive and increases societal welfare. Wang (2018) utilized Chinese microenterprise data from 2000 to 2006 to examine the impact of trade policy uncertainty on the earnings of export companies. In both the short and long term, the decline in trade policy uncertainty increased the revenues of exporting companies. Mao (2020) found that uncertainty over trade policy affects corporate savings.

**H3.** *Trade policy uncertainty (TPU) and research and development (R&D) significantly affect patents (innovation) under the VAR and granger causality models.*

## 3. Economic Framework

This section provides an introduction to economic theory and a discussion of the state's decision to invest in new technology. By using the work of Handley and Limão (2017) as a basis, we focus on the decision of the nation and enterprise to invest in innovation to draw vital conclusions. Monopolistic rivalry represents the only specialized sector of the economy. The output of businesses can be increased by making a payment on a non-recoverable investment (Bustos 2011).

### 3.1. Theoretical Background

We consider a configuration with two countries: the domestic nation and the international nation. Here, $n$ represents the country, with $d$ representing the domestic and $x$ representing the international country. Conveniently, there is just one differentiated sector $j$ with monopolistic competition and one type $i$. The productivity of businesses varies, as assessed by $i$. Investing in modern technology can increase the efficiency of the industry. R&D investments incur buried costs, including expenses such as acquiring specific assets, hiring or training specialized personnel, acquiring information on new technologies, etc. Investing in R&D yields superior technology, which reduces the marginal cost of manufacturing from A to B; failing to spend yields inferior technology and initial productivity $i0$.

For an industry producing variety, we face an ad valorem tariff $Tx = x1$. Because the industry is subject to the same tariffs, there are no fixed costs of entering a foreign market. Thus, every domestic company exports to worldwide markets. Lastly, there is an exogenous likelihood of exit 1 in every era, independent of business productivity. The sum of domestic and export revenues determines equilibrium per-period operating profits and profits from low-tech production are

$$\pi(\varphi_{i0}) = \pi_d(\varphi_{i0}) + \pi_x(\varphi_{i0}) = B_d \varphi_{i0}^{\sigma-1} + B_x \tau_x^{-\sigma} \varphi_{i0}^{\sigma-1}. \tag{1}$$

We received the following profit, after investment in research and development:

$$\pi(\varphi_{i1}) = \pi_d(\varphi_{i1}) + \pi_x(\varphi_{i1}) = B_d \varphi_{i1}^{\sigma-1} + B_x \tau_x^{-\sigma} \varphi_{i1}^{\sigma-1}. \tag{2}$$

*3.2. Innovation and Uncertainty Decision*

Here, we examine the case of a domestic corporation that can spend on R&D to improve productivity but is uncertain about potential overseas market circumstances. A broader market makes R&D more profitable. However, future international market access is uncertain due to changing trade policies as $T = 1$. There is uncertainty over applicable foreign tariffs. Period $t$ presents a binary choice for the firm: invest in R&D now or wait until period $t$. The only unknowns are the external survival rate. Investment in R&D yields a constant stream of domestic and export earnings due to productive output technologies,

$$\Pi^I(\tau_s, \varphi_1) = \Pi_d^I(\varphi_1) + \Pi_x^I(\tau_s, \varphi_1). \tag{3}$$

Without time discounting, expected domestic profits are as given below:

$$\Pi_d^I(\varphi_1) = \pi_d(\varphi_1) + \sum_{t=1}^{\infty} \beta^t \pi_d(\varphi_1) = \frac{\pi_d(\varphi_1)}{1 - \beta}. \tag{4}$$

Furthermore, potential export profits are presented through the following equation,

$$\Pi_x^I(\tau_s, \varphi_1) = \pi_x(\tau_s, \varphi_1) + \sum_{t=1}^{\infty} \beta^t \pi_x(\tau_s', \varphi_1), \tag{5}$$

where trade policy information refers to $s$, and the firm's productivity utilizing high-type technology are all used to calculate $es$, an estimate of the future worth of $b$. The expected value of a firm without up-gradation and export profit has been obtained by using the standard technology:

$$\Pi(\tau_s, \varphi_0) = \Pi_d(\varphi_0) + \Pi_d(\tau_s, \varphi_0). \tag{6}$$

The anticipated domestic profit is given as

$$\Pi_d(\varphi_0) = \pi_d(\varphi_0) + \sum_{t=1}^{\infty} \beta^t \pi_d(\varphi_0) = \frac{\pi_d(\varphi_0)}{1 - \beta}. \tag{7}$$

After that, the anticipated export profit is given as

$$\Pi_x(\tau_s, \varphi_0) = \pi_x(\tau_s, \varphi_0) + \mathbb{E}_x \sum_{t=1}^{\infty} \beta^t \pi_x(\tau_s', \varphi_0). \tag{8}$$

Now, we discuss the case for comprehensive explanation. As proven by $f$, investing is advantageous if there is no uncertainty regarding future market access. When the estimated value of investing net of sunk investment cost exceeds the estimated value of producing with low-type technology, it refers to investment, and there is no value in waiting. This is the state of investing apathy:

$$\left[ \pi_d(\varphi_1) - \pi_d(\varphi_0) \right] + \left[ \pi_x(\tau_s^D, \varphi_1) - \pi_x(\tau_s^D, \varphi_0) \right] = I \, (1 - \beta). \tag{9}$$

In place of that, the industry must either spend now or continue manufacturing low-tech equipment while conditions improve. For this flexible investment decision, an optimal stopping issue represents an investment by stopping and waiting by prolongation. Bellman's equation is utilized to address a company's choice dilemma, as given through the following equation:

$$F(\tau_s, \varphi) = \max\left\{\Pi_d^I(\varphi_1) - \Pi_d(\varphi_0) + \Pi_d^I(\tau_s, \varphi_1) - \Pi_x(\tau_s, \varphi_0) - I = \beta\mathbb{E}_x F(\tau'_s, \varphi)\right\}. \tag{10}$$

In this optimal stopping issue, the range of $T$ is divided into a continuation and stopping regions, respectively. As a general rule, ideal intervals for termination can alternate with optimal intervals for continuation. Under plausible hypotheses, it is possible to prove that a single threshold value of $T$ and $R$ generates a clean separation of the range of $T$ into a continuation region and stopping area, respectively:

$$\Pi_d^I(\varphi_1) - \Pi_d(\varphi_0) + \Pi_d^I(\tau_s^U, \varphi_1) - \Pi_x(\tau_s^U, \varphi_0) - I = \beta\mathbb{E}_x F(\tau_s'^U, \varphi)\right\}. \tag{11}$$

Under uncertainty, the condition of investment indifference presented through the following equation:

$$F(\tau_s^U, \varphi) = \Pi_d^I(\varphi_1) - \Pi_d(\varphi_0) + \Pi_x^I(\tau_s^U, \varphi_1) - \Pi_x(\tau_s'^U, \varphi_0) - I \tag{12}$$

To analyze the rule of uncertainty, it is helpful to rearrange Equation (10) by subtracting Equation (12), as given below:

$$V_s = max\left\{0, \ \mathrm{B}\mathbb{E}_x V'_s - [\pi_d(\varphi_1) - \pi_d(\varphi_0)] - [\pi_x(\tau_s, \varphi_1) - \pi_x(\tau_s, \varphi_0)] + (1 - \beta)I\right\}. \tag{13}$$

*3.3. Trade Policy Regime*

This policy regime is modeled around trade policy, as introduced by Handley and Limão (2017). A Markov chain is used to explain the three possible trade policy states, denoted by $s = 0$, 1, and 2. There are three different levels of trade protection: the highest level of protection, 2, has duties that are twice as high as the lowest level, 0, which has zero tariffs. Trade policy uncertainty only exists in the intermediate state ($s > 0$) and is considered non-absorbent under severe circumstances,

$$S = \begin{bmatrix} \lambda_{00} & 0 & 0 \\ \lambda_{10} & \lambda_{11} & \lambda_{12} \\ 0 & 0 & \lambda_{22} \end{bmatrix}, \tag{14}$$

where $\lambda_{11} = (1 - \gamma)$, $\lambda_{12} = \gamma\lambda$, and $\lambda_{10} = \gamma(1 - \lambda)$ are the numbers. A trade policy shock affects all firms equally because they all have the same ideas and are exposed to the same risk.

## 4. Data and Methodology

*4.1. Population and Sample*

The world population review covered the sample of 65 developing nations and provided annual data from 1980–2020. This analysis focuses on the period following 1980 for two reasons. Between 1980 and 2020, the majority of nations joined the WTO. The first nation joined WTO in 1995 and the last in 2016. The statistics of each country will be divided into pre-and post-accession years. Multiple factors diminished policy unpredictability. The PNTR went into effect when the nation joined the WTO. Following Pierce and Schott (2016), we consider each country's post-PNTR WTO membership years. Secondly, we chose this sample to study trade policy changes during the 2008 recession. This financial crisis hit almost all nations, most recently due to COVID-19. From 2019 to 2020, the pandemic's hottest time rendered trading impossible, yet medical innovation, research, and trade surged. Aid for medical equipment, vaccines, and research affected free trade policy among countries.

This field study assists underdeveloped nations. Table 1 describes each variable's capacity, source, and predicted impact on a country's trade policy uncertainty and innovation.

**Table 1.** Data variables, description and expected impact.

| Variable | Capacity | Description | Source | Duration | Exp Sign |
|---|---|---|---|---|---|
| Patent | Dependent Variable | A government authority or license grants a right or title for a limited time, notably the exclusive right to restrict others from manufacturing, to utilize, or selling an invention. | WIPO | 1980–2020 | (+) |
| $TPU = \ln\left(\frac{t2}{t1}\right)$ | Independent Variable | TPU is a weighted average log difference between "column 2" and MFN tariffs. | TRAINS | 1980–2020 | (+) |
| R&D | Independent Variable | The money is spent on systematic creative labor to enhance the stock of knowledge and use this information to create new applications. | WDI | 1980–2020 | (+) |
| FDI | Control Variable | A foreign investment in the form of a controlling stake in a business in one country by a foreign corporation. | WDI | 1980–2020 | (+) |
| NTM | ControlVariable | Non-tariff barriers (NTBs)/Non-tariff measures (NTMs) are trade restrictions restricting imports or exports of products or services through means other than tariffs. | TRAINS | 1980–2020 | (+) |
| Imports | Control Variable | The value of all commodities and other market services received from the rest of the world is represented by imports. | WDI | 1980–2020 | (+) |
| Import Tariff | Control Variable | Also known as customs duty, the tax imposed on commodities as they cross national borders, typically by the country's government doing the importing. | TRAINS | 1980–2020 | (–) |

**Note:** The data for the variable discussed in this table are derived from the reliable sources of WDI, WIPO, and TRAINS from 1980 to 2020 on an annual basis. The values assigned for respective variables are patents as innovation (PAT), trade policy uncertainty (TPU), research and development expenditures (R&D), foreign direct investment (FDI), country's imports (Imp), imports tariff (ImpT), and non-tariff measures (NTM).

### 4.2. Econometric Modeling

The empirical strategy uses time–country variations based on the generalized double difference-in-differences technique. Because the sector is constant, the sector difference is not considered. Comparing industries with greater and less vulnerability to trade policy uncertainty reveals similar patenting and innovation tendencies. A difference-in-differences technique can discover the causal relationship between trade policy uncertainty and innovation, assuming this assumption is true (Coelli 2018). These are the empirical models developed by using the approach outlined above.

#### 4.2.1. Difference-in-Differences (DID) Method

We have estimated the following generalized triple difference-in-differences model under the scope of this study:

$$Ln\,(p_{jnt}) = \alpha + \omega_{nt} + \omega_{jt} + \delta1\,\ln\,(TPU) + \delta2\,lnR\&D + \mu. \tag{15a}$$

After that, we included the post PNTR as interaction term with TPU, which helps to generate the next model, as given below,

$$Ln\ (p_{jnt}) = \alpha + \omega_{nt} + \omega_{jt} + \delta 1 postPNTR_t * ln\ (TPU) + \delta 2\ lnR\&D + \mu, \tag{15b}$$

where $ln\ (p_{jnt})$ is the dependent variable as log number of granted invention patents by all applicants in nation $n$ in technology $j$ and year $t$. The patent's four-digit technology code experimentally identifies the technology $j$, and $n$ refers to the nation of residence of the applicant (patentee), not the patent office where the patent application was submitted. The dummies $nt$ and $jt$ stand for country time, country technology, and technology time, respectively. Post-PNTR dummy (Post-PNTRt), the trade policy uncertainty exposure (lnTPU), and the term of interest make up the fourth term on the right. The uncertainty exposure measure (lnTPU) is a weighted average of log differences between column 2 tariffs and MFN tariffs, as measured through the following formula:

$$Ln\ (TPU_j) = \sum_h w_{jh} ln\left(\frac{t2}{t1}\right).$$

According to the researchers' interpretation, this weight represents either the relative relevance of each high-tech HS product that can be created by using technology j or the researcher's uncertainty when mapping an unpatented technique to an unpatented product. It is supplied by Lybbert and Zolas (2012), who illustrate the correspondence between patent technology (IPC) and product (HS) codes.

### 4.2.2. Long-Run and Short-Run Effect

The panel autoregressive distributed lag (ARDL) bound approach is used to determine the long-term and short-term effects of TPU and R&D expenditure on innovation in the medical sector of developed and developing nations,

$$Y_{it} = \alpha + \alpha 1 X_{it} + \beta_1 Y_{i,t-1} + \beta_2 X_{i,t-1} + \varepsilon_{it}. \tag{16}$$

The model, as presented through Equation (16), is the general form of the ARDL model. The panel ARDL technique was chosen to examine long- and short-term co-integration correlations between determinants and extract the error correction version (ECM) of the panel properties to discover short-term dynamics. In addition, alternative co-integration techniques, such as the Johansen and Juselius (1990) and traditional Johansen (1995) techniques, achieved comparable results. However, the panel autoregressive distributed lag approach was chosen over co-integration because of its added benefits, as defined below:

$$Dln\ (p_{jnt})_t = \alpha + \sum_{i=1}^{k} d\delta 1 ln(TPU)_{t-1} + \sum_{i=1}^{k} d\delta 2\ lnRandD_{t-1} + \Phi 1\ ln(TPU)_{t-1} + \Phi 2\ lnR\&D_{t-1} + \mu. \tag{17}$$

No co-integration assumption can be tested by using the F test, which does not have a usual allocation based on (1) whether the elements involved in the model are entirely I (0), fully I (1), or a combination of both, (2) the number of estimators, and (3) if there is a trend, intercept, or both. Hammar and Belarbi (2021) also proposed a future study to analyze developing nations to resolve the endogeneity issue.

### 4.2.3. Panel VAR Model

Panel VAR has rechecked the reverse causality issue on the medical sector's respective indicators. Numerous investigations on the connection between innovation and global exchange also contain innovative worldwide exchange's impacts. Coe and Helpman (1995) and Coe et al. (1997) investigated the manners in which how the global exchange advanced mechanical advancement. In this regard, the panel vector autoregressive model

examines the relationships between trade policy, innovation, and R&D expenditures among developed and developing economies, as given through the following equation:

$$lnp_{it} = \Phi + \sum_{K=1}^{P} \Phi_{1k} ln(TPU)_{it-k} + \sum_{K=1}^{P} \Phi_{2k} R\&D_{it-k} + \varepsilon_{it} . \tag{18}$$

The autoregressive distributed lag approach treats all variables as dependent and independent. A further argument for employing the PVAR model is that, when using panel data, the temporal variation is more substantial than in cross-sections, and there is a direct relationship between all policy uncertainty variables.

### 4.3. Data Estimation Method

Patents in the medical sector were examined in both developed and developing countries with the perspective of Snow (1855), which was a controlled before-and-after study in various social sciences, whereas the fast-food industry in New Jersey and Pennsylvania was studied by Card and Krueger (1993). The DID estimate and labor market regulations were also studied when Meyer et al. (1995) examined workers' compensation and absenteeism.

The DID focuses on serially associated outcomes while ignoring standard error inconsistency. Bertrand et al. (2003) highlight possible DID prejudice and documented the three ways to overcome prejudices. When there are enough groups, block bootstrap can compute consistent standard errors. Even with a limited number of states, compressing data produces consistent standard errors (though the power of this test declines fast). When the number of groups is large enough, autocorrelation can be used to calculate standard errors. Therefore, we separated the data into pre-and post-time periods to eliminate error biases. The variance inflation factor (VIF) approach was used to test multicollinearity. Levine (1997) found no connection between explanatory factors when VIF = 1, but multicollinearity when VIF > 10. Multicollinearity was indicated when the VIF exceeded 5 (Gujarati and Porter 2009).

Then, estimates and inference regarding the model's long-run properties are performed as suggested by Hendry et al. (1984) and Wickens and Breusch (1988). Different stationary variables, often called I (1) variables, complicate the study. Co-integration literature analyzes long-run correlations between I (1) variables, and its essential premise is that the classic ARDL approach is invalid in the presence of I (1) variables. A list of alternative estimation and hypothesis testing strategies for I (1) variables has been conducted, following Phillips and Loretan (1991). Pesaran et al. (1999) investigated the ARDL modeling approach for co-integration analysis to tackle this technique's difficulty. Johansen and Juselius (1990) and Pesaran et al. (1999) explored long-run structural modeling in the framework of an unrestricted VAR model. Applying the ARDL method in this study to systems with short-term and long-term identifying restrictions would bring us back to the Cowles commission approach presented in panel ARDL (Im and Pesaran 2003; Westerlund 2007). Coe et al. (1997) employ the PVAR model to assess the level of dynamic heterogeneity and reverse causality in the trade sector of economics.

More precisely, the panel ARDL model is used for the long- and short-run impact of independent variables on dependent variables in case the variables are stationary at I (0) or integrated of I (1). PVAR model has been implied to check the effect and shocks of each variable on another variable. The IV regression model has multiple reasons to apply, such as checking the robustness and potential endogeneity issues in the model.

## 5. Empirical Results
### 5.1. Descriptive Statistics

Table 2 offers descriptive statistics for developing countries' mean, standard deviation, minimum, and maximum values. Patent and trade policy uncertainty drive R&D spending. Throughout the study period, every developing country filed 5.988 innovation patents annually. The mean TPU is 1.054, and the average R&D spending is 0.372. This table also shows other control variables for the model's robustness assessment.

**Table 2.** Descriptive statistics.

| Variables | Obs | Mean | Std.Dev. | Min | Max |
|---|---|---|---|---|---|
| PAT | 2665 | 5.988 | 2.423 | 0.000 | 14.267 |
| TPU | 2665 | 1.054 | 2.496 | −71.935 | 61.781 |
| R&D | 2665 | 0.372 | 1.228 | −9.854 | 12.124 |
| Imp | 2665 | 1.512 | 0.279 | −0.365 | 2.622 |
| FDI | 2665 | 0.719 | 4.984 | −39.428 | 44.010 |
| ImpT | 2665 | 10.472 | 12.123 | −42.730 | 148.650 |
| NTM | 2665 | 36.465 | 26.195 | 0.500 | 95.170 |

**Note:** This table covers the descriptive statistics of all selected explained and explanatory variables for 65 developing countries from 1980 to 2020, allowing for 2665 observations.

### 5.2. Correlation Matrix

Table 3 illustrates the study's relationship between dependent and independent variables through correlation analysis. Trade policy uncertainty, imports, and import tariff coefficient values of 0.0216, 0.159, and 0.027, respectively, have a positive confidence interval of 99% when applied to medical innovation. There is also a correlation of 0.064 and 0.027 between R&D and TPU, as well as a correlation of −0.112 for imports tariffs (ImpT) and 0.047 for imports (Imp), according to R&D (Shen and Hou 2021). Non-trade measures (NTMs) are essential for trade control, as they are an essential component. TPU and imports correlate adversely with NTM, whereas R&D spending and FDI correlate positively.

**Table 3.** Correlation matrix.

| Variables | PAT | TPU | R&D | Imp | FDI | ImpT | NTM |
|---|---|---|---|---|---|---|---|
| PAT | 1.000 | | | | | | |
| TPU | 0.0216 *** | 1.000 | | | | | |
| R&D | 0.027 * | 0.064 *** | 1.000 | | | | |
| Imp | −0.159 *** | −0.0419 ** | 0.047 ** | 1.000 | | | |
| FDI | 0.113 *** | 0.094 *** | −0.0009 | 0.172 *** | 1.000 | | |
| ImpT | −0.027 | −0.012 | −0.112 *** | −0.294 *** | −0.041 ** | 1.000 | |
| NTM | 0.275 *** | −0.045 ** | 0.096 *** | −0.214 *** | 0.076 *** | −0.016 | 1.000 |

**Note:** This table presents the correlation matrix among the dependent, independent, instrumental, and control variables. It shows the direction of variables in which they are related. The table shows the correlation between the patents as innovation (PAT), trade policy uncertainty (TPU), research and development expenditures (R&D), foreign direct investment (FDI), country's imports (Imp), import tariff (ImpT), and non-tariff measures (NTM). *** $p < 0.01$, ** $p < 0.05$, * $p < 0.1$.

### 5.3. Multicollinearity Diagnostic Test

Table 4 uses VIF to test for multicollinearity in multivariate models. Melo and Kibria (2020) define multicollinearity as the absence of correlation between explanatory variables when the VIF equals one. Table 4 documents a non-multicollinearity in the study's explanatory components. In the absence of collinearity, panel regression models are more reliable and consistent because multicollinearity reduces their statistical power.

**Table 4.** Multicollinearity diagnostic test.

| Models | | TPU | R&D | Imp | FDI | ImpT | NTM | MeanVIF |
|---|---|---|---|---|---|---|---|---|
| 1 | VIF | 1 | 0.9998 | | | | | 1.000 |
| | 1/VIF | 1 | 0.9998 | | | | | |
| 2 | VIF | 1.006 | 1.013 | 1.128 | 1.036 | 1.107 | | 1.058 |
| | 1/VIF | 0.994 | 0.987 | 0.886 | 0.966 | 0.904 | | |
| 3 | VIF | 1.006 | 1.024 | 1.203 | 1.051 | 1.113 | 1.082 | 1.08 |
| | 1/VIF | 0.994 | 0.977 | 0.832 | 0.952 | 0.898 | 0.924 | |

**Note:** A multicollinearity test was performed on each model based on equations 5 to 8, and the VIF test statistics are presented above.

### 5.4. Does More Patenting Mean More Innovation?

　　　Table 5 presents the baseline results of the generalized DID estimation method among the innovation, TPU and R&D expenditures with control variables using the time and country fixed effect. Column 1 of this table shows the TPU estimates with fixed effect, which compares sectors with high and low exposure to policy uncertainty. For the developing group of countries, it has been shown that there is an inverse correlation between innovation and trade policy uncertainty. As mentioned in the model, innovation is dependent and TPU is the independent variable, so the correlation effect is going from TPU to innovation. Secondly, TPU or policies are the former ones that impact the latter one like innovation, which is dependent on the sustainability of the trade policy uncertainty. Whereas csolumns 2 to 6 provide the outcomes of R&D expenditures toward innovation, it contains the controls for WTO-related policy changes that were implemented at the same time. Indicators of FDI restrictions, non-tariff barriers, import duties, and imports are all included in these measures. TPU has a detrimental, time-fixed effect on innovation because these outcomes are for low-income nations with uncertain policies (Borojo et al. 2022).

　　　Patenting is often found to be associated with R&D and other indicators of innovation (Griliches 1990). In column 2, findings show that countries' R&D expenditures benefit innovation, allowing them to increase patents by increasing R&D spending (Dong et al. 2021). For further robustness, we have selected the control variables as FDI, imports, import tariff, and NTM, presented in columns 3 to 6 in this table. We have documented that FDI, imports, and NTM all favor innovation, but import tariffs have a negative impact. The negative association was observed because import tariffs are the main indicator to control the inflow of goods in response to demand and the inflow of patented and unpatented innovation through trade, as consistent with Coelli (2018). As intended, the coefficient of TPU is positive and statistically significant, indicating that policy uncertainty ex-ante is connected with enhanced innovation once trade policy uncertainty has been eradicated. As per previous work on these indicators, most of them are related to the one or two developed countries, which are now at the stage of sustaining innovation, instead of working on the fulfillment of necessities or the basics of innovation and trade-like developing countries. Any uncertainty in the world market has a huge impact on developing and low-income countries as compared to high-income countries, as discussed by Borojo et al. (2022).

**Table 5.** Difference in differences (DID) baseline results.

| Variables | 1 | 2 | 3 | 4 | 5 | 6 |
|---|---|---|---|---|---|---|
| TPU | −0.0186 *** | −0.0179 *** | −0.0181 *** | −0.0155 *** | −0.0160 *** | −0.0061 *** |
| | (0.00154) | (0.00153) | (0.00154) | (0.00153) | (0.00152) | (0.00852) |
| R&D | | 0.114 *** | 0.113 *** | 0.124 *** | 0.125 *** | 0.0890 *** |
| | | (0.00302) | (0.00304) | (0.00305) | (0.00302) | (0.0206) |
| FDI | | | 0.00176 * | −0.00447 *** | −0.00525 *** | 0.0253 *** |
| | | | (0.00105) | (0.00107) | (0.00106) | (0.00618) |
| Imp | | | | 0.597 *** | 0.594 *** | 0.493 *** |
| | | | | (0.0210) | (0.0208) | (0.123) |
| ImpT | | | | | −0.0213 *** | −0.0168 *** |
| | | | | | (0.000519) | (0.00239) |
| NTM | | | | | | 0.0259 *** |
| | | | | | | (0.00921) |
| Observations | 87,945 | 87,945 | 87,084 | 87,084 | 87,084 | 87,084 |
| $R^2$ | 0.031 | 0.046 | 0.046 | 0.054 | 0.073 | 0.085 |
| Fixed-Effect | yes | yes | Yes | yes | yes | yes |

**Note:** This table reports the double difference-in-difference estimate in column 1 and the generalized double difference in difference in all models. The independent variable is the TPU. Constant, country time, and country technology fixed effect has been included but not reported here. Additional controls have been included regarding the country's NTM, imports from another world, FDI restrictions, and import tariff data. The data spam is from 1980 to 2020. Standard errors in parentheses. *** $p < 0.01$, * $p < 0.1$.

Before the 2001 PNTR conference, trade policy uncertainty was unrelated to innovation. Table 6 displays the results for post-PNTR time events by using the generalized DID estimation method among the innovation, TPU, and R&D expenditures with control variables by using the time and country fixed effect, in which the post-PNTR dummy is replaced in the main model and interacted with all variables. Columns 2 to 6 of this table evaluate the DID estimates of post-PNTR analysis, which contains country time and country technology fixed effect. Columns 2 to 6 provide the outcomes of control variables integrated with the PNTR dummy factor, which allows us to analyze the events of innovation and TPU in the post-PNTR period.

**Table 6.** Difference in difference (DID) post-PNTR results.

| Variables | 1 | 2 | 3 | 4 | 5 | 6 |
|---|---|---|---|---|---|---|
| P.TPU | −0.0197 *** | −0.0221 *** | −0.0164 *** | −0.0176 *** | −0.0192 *** | −0.0208 *** |
| | (0.00292) | (0.00293) | (0.00299) | (0.00299) | (0.00299) | (0.00299) |
| P.R&D | | 0.103 *** | 0.138 *** | 0.151 *** | 0.114 *** | 0.115 *** |
| | | (0.0117) | (0.0122) | (0.0123) | (0.0125) | (0.0125) |
| P.Imp | | | −0.121 *** | −0.194 *** | −0.247 *** | −0.232 *** |
| | | | (0.0129) | (0.0138) | (0.0142) | (0.0143) |
| P.ImpT | | | | 0.0130 *** | 0.0118 *** | 0.0115 *** |
| | | | | (0.000901) | (0.000902) | (0.000903) |
| P.NTM | | | | | 0.00519 *** | 0.00500 *** |
| | | | | | (0.000312) | (0.000313) |
| P.FDI | | | | | | 0.0174 *** |
| | | | | | | (0.00213) |
| Observations | 87,945 | 87,945 | 87,945 | 87,945 | 87,945 | 87,945 |
| $R^2$ | 0.029 | 0.030 | 0.031 | 0.034 | 0.037 | 0.037 |
| Fixed-Effect | yes | Yes | yes | yes | yes | yes |

**Note:** This table reports the double difference-in-difference estimate in column (1) and the generalized double difference in difference in all models. The independent variable is the interaction term of the post-NTR dummy and TPU. Constant, country time, and country technology fixed effect has been included but not reported here. Additional controls have been included regarding the country's NTM, imports from another country, FDI restrictions, and import tariff data. The data spam is from 1980 to 2020. Standard errors in parentheses, (*** $p < 0.01$).

The findings in column 1 of this table reported that a significant and negative relationship exists between innovation and TPU, which leads to the conclusion that a change in pre-WTO TPU exposure leads to a negative change in post-PNTR patenting activity. Moreover, findings in column 2 show that R&D has a significant and positive impact on innovation, showing that pre-WTO spending through R&D directly affected patenting after the PNTR period, as consistent with Borojo et al. (2022) and Dong et al. (2021).

It would appear that the United States is the largest export market for health-related technologies; however, there are a number of organizations and categories for developing countries that do not include either the United States or European nations; this includes many in Asia, Africa, etc. Furthermore, the United States does not participate in trade between these countries, despite the fact that they provide cheap goods to each other through bilateral trade. Foreign direct investment (FDI) from the United States facilitates the export of certain technologies to emerging economies. China, albeit still considered a developing country, has a substantial export market in the medical field; hence it was only products from other developing countries that were considered for this study.

According to results of Tables 5 and 6, the impact of R&D investment is greater in the post-PNTR period as opposed to the entire time period, which is consistent with the expectations regarding investment and innovation following the post-PNTR.

Table 7 demonstrates the results of two-stage least square (2SLS) estimation by using post-PNTR *lnT2 as an instrument for post-PNTR *lnTPU with country time and country technology fixed effect. Results for 2SLS estimations are shown in different steps. First, OLS estimation has been run, then reduced from, first stage and 2SLS, because it is possible

to instrument the baseline uncertainty exposure metric by using column 2 tariffs from Smoot–Hawley. As Coelli (2018) discussed, the uncertainty exposure meter, lnTPU, is plausibly exogenous because virtually all fluctuation derives from the 1930 column 2 tariffs introduced by Smoot–Hawley. If the U.S. purposely applied MFN tariffs, this would result in smaller log differences between column 2 and MFN tariffs, biasing the findings against identifying an uncertainty effect on innovation.

The results show that for both OLS and 2sls, all variables are significant, but P.TPU, PFDI, and P.impT have a negative impact and PRD has a positive impact on innovation. In reduced form equation estimations and first stage estimation, the instrumental variable column 2 tariff is highly significant but negatively impacts innovation in developing countries. The results concluded that the estimated effect is statistically significant and equivalent in magnitude to the baseline estimation.

**Table 7.** IV Estimates.

| Variables | OLS | RF | 2SLS | FS |
|---|---|---|---|---|
| P.TPU | −0.0687 ** | | −0.0726 ** | |
| | (0.0318) | | (0.0325) | |
| P.R&D | 1.368 *** | 1.166 *** | 1.342 *** | 1.302 *** |
| | (0.128) | (0.123) | (0.131) | (0.127) |
| P.NTM | 0.0157 *** | 0.0124 *** | 0.0170 *** | 0.0149 *** |
| | (0.00205) | (0.0020) | (0.00212) | (0.00204) |
| P.FDI | −0.0425 *** | −0.0555 *** | −0.0545 *** | −0.0478 *** |
| | (0.00914) | (0.0090) | (0.00980) | (0.00915) |
| P.ImpT | −0.0213 *** | −0.0231 ** | −0.0186 ** | −0.0184 ** |
| | (0.00784) | (0.0076) | (0.00805) | (0.00781) |
| $\ln\tau_{col2}$ | | −0.4325 *** | | −0.570 *** |
| | | (0.1086) | | (0.113) |
| Observations | 2665 | 2665 | 2665 | 2665 |
| $R^2$ | 0.099 | 0.0956 | 0.050 | 0.107 |

**Note:** This table indicates 2sls generalized double difference in difference estimates of inverse hyperbolic sine of patents and trade policy uncertainty. TPU is estimated with the instrumental variable column 2 tariff. Constant, country time, and country technology fixed effect has been included but not reported here. Column (1–4) reports the OLS, first stage (FS), 2sls, and reduced form (RF) estimates. Additional controls have been included regarding the country's NTM, imports from another world, FDI restrictions, and import tariff data. The data spam is from 1980 to 2020. Standard errors in parentheses. *** $p < 0.01$, ** $p < 0.05$.

*5.5. IV Regression Estimates*

Table 8 discusses the results of the panel ARDL model without control variable and control variable for the long-run and short-run impact of TPU and R&D on innovation. For each model, long-run and short-run estimation results are separated. In model 1, the long-term estimation findings indicate that the TPU and R&D computed factors for long-term innovation fluctuation are statistically significant in both the negative and positive directions, showing that medical innovation in developing nations is increasing. In other words, the outcomes correspond to our expectations. TPU's effects on the nations of a destination significantly impact the vast array of potential new ideas (Borojo et al. 2022). Short-term estimation results indicate that TPU and R&D variations have no substantial effect on innovation; however, the significant EC term indicates long-term co-integration. There is a substantial disparity between the estimation results for both terms, indicating that uncertainty and R&D expenditures affect innovation positively and negatively only when long-term investment plans are implemented.

**Table 8.** Long-run and short-run estimates (ARDL).

| Variables | 1 | 2 |
|:---:|:---:|:---:|
| **Panel A—Long-Run Estimates** | | |
| TPU | −0.514 *** | −0.269 *** |
| | (0.0805) | (0.0639) |
| R&D | 0.116 *** | 0.100 *** |
| | (0.0227) | (0.0225) |
| FDI | | 0.0351 ** |
| | | (0.0144) |
| Imp | | 2.629 *** |
| | | (0.258) |
| ImpT | | −0.00638 *** |
| | | (0.00164) |
| **Panel B—Short-Run Estimates** | | |
| EC | −0.190 *** | −0.241 *** |
| | (0.0284) | (0.0296) |
| D.TPU | −0.0272 | −0.113 |
| | (0.0802) | (0.104) |
| D.R&D | 0.130 | 0.00953 |
| | (0.356) | (0.352) |
| D.FDI | | −0.0508 |
| | | (0.0338) |
| D.Imp | | −0.218 |
| | | (0.285) |
| D.ImpT | | −0.0236 |
| | | (0.0168) |
| Observations | 2600 | 2560 |

**Note:** This table evaluates the long- and short-run analysis by using the panel ARDL model. In column 1, the impact of independent variables TPU and R&D expenditures has been estimated for the long-run and short-run relationship. In column 2, the other controls, FDI, import, and import tariff, have been included for the long-run and short-run fluctuations. The data spam is from 1980 to 2020. Panel A discussed the long-run estimates, and Panel B covers the short-run impact. Standard errors in parentheses, (*** $p < 0.01$, ** $p < 0.05$).

### 5.6. Long-Run and Short-Run Estimates (ARDL)

In model 2 of Table 8, when control variables are added to evaluate model robustness, long-run estimation results indicate that the TPU, R&D, and control variables computed components for long-term innovation fluctuations are statistically significant. Short-term estimate results indicate that variations in TPU, R&D, and control variables have little effect on innovation, whereas the many EC terms indicate long-run co-integration. There is a substantial disparity between the estimation results for both terms, indicating that uncertainty and R&D expenditures affect innovation positively and negatively only when long-term investment plans are executed, as opposed to short-term investment plans (Altıntaş and Kassouri 2020; Bottazzi and Peri 2007; Koc and Bulus 2020).

### 5.7. Panel Vector Autoregressive (PVAR) Model Estimates

5.7.1. Lag Length Criteria

Lag determination statistics for developing nations are shown in Table 9. Akaike information criteria (AIC) and Hannan–Quinn information criteria (HQ), and Schwarz information criteria (SC) are used to determine the lag. Four lags in underdeveloped countries are considered in the study.

**Table 9.** Lag selection for PVAR model estimation.

| Lag | LogL | LR | FPE | AIC | SC | HQ |
|---|---|---|---|---|---|---|
| 0 | −13,252.52 | NA | 12.30087 | 11.02330 | 11.03052 | 11.02593 |
| 1 | −5423.474 | 15632.05 | 0.018435 | 4.520145 | 4.549011 | 4.530645 |
| 2 | −5234.377 | 377.0923 | 0.015871 | 4.370376 | 4.420892 | 4.388752 |
| 3 | −5103.810 | 260.0482 | 0.014345 | 4.269281 | 4.341447 | 4.295532 |
| 4 | −5026.495 | 153.7940 * | 0.013553 * | 4.212470 * | 4.306286 * | 4.246597 * |

**Note:** This table compares developed and developing countries' lag determination statistics. Akaike information criteria (AIC) and Hannan–Quinn information criteria (HQ), and Schwarz information criteria are used to determine the lag (SC). The study uses four lags for developing countries based on AIC, HQ, and SC. * $p < 0.1$.

5.7.2. Panel Vector Autoregressive (PVAR) Estimates

Table 10 presents the results of the panel VAR model of innovation, TPU, and R&D expenditures to capture the joint dynamics of multiple time series by considering all variables as endogenous. The estimated results show that all factors are significantly related to their lags but not correlated to the lags of other variables. It demonstrated that a country's current innovation and R&D spending significantly and favorably impact developing countries; therefore, TPU, innovation, R&D expenditures, and their corresponding lags are found to have reverse causality. TPU is the factor that creates uncertainty, and because of this, it has a significant but negative impact on its previous year's value but a good impact on the fourth lag. R&D is the factor that justifies the investment in innovation, which is why it has a positive and significant impact on its previous year's value, but a negative impact on fourth lags. In addition, PAT is the factor that also depends on its previous value, as it has a significant and positive relationship with its previous three lags. These results concluded that the lags in innovation, TPU, and R&D are linked to the contemporary reversal causality impact on TPU, (Yan et al. 2022).

**Table 10.** Panel vector autoregressive (PVAR) estimates.

| | LPAT | TPU | R&D |
|---|---|---|---|
| LPAT (−1) | 0.687825 *** | −0.052735 | −0.002889 |
| | (0.02237) | (0.05761) | (0.00447) |
| LPAT (−2) | 0.208514 *** | 0.024926 | −0.000912 |
| | (0.02731) | (0.07033) | (0.00545) |
| LPAT (−3) | 0.060634 ** | −0.046649 | 0.003050 |
| | (0.02726) | (0.07019) | (0.00544) |
| LPAT (−4) | 0.021892 | 0.084311 | 0.002921 |
| | (0.02270) | (0.05847) | (0.00453) |
| TPU (−1) | 0.000142 | −0.041015 *** | −0.000161 |
| | (0.00632) | (0.01627) | (0.00126) |
| TPU (−2) | −0.001971 | −0.002292 | −0.000577 |
| | (0.00628) | (0.01618) | (0.00126) |
| TPU (−3) | −0.003510 | −0.021917 | $5.77 \times 10^{-5}$ |
| | (0.00623) | (0.01604) | (0.00124) |
| TPU (−4) | −0.003739 | 0.051685 *** | −0.000102 |
| | (0.00488) | (0.01256) | (0.00097) |
| R&D (−1) | 0.024515 | 0.125335 | 1.075814 *** |
| | (0.10088) | (0.25981) | (0.02015) |
| R&D (−2) | −0.088851 | 0.069969 | 0.143930 *** |
| | (0.14937) | (0.38468) | (0.02983) |
| R&D (−3) | 0.119517 | −0.100749 | −0.029939 |
| | (0.14923) | (0.38432) | (0.02980) |
| R&D (−4) | −0.049766 | −0.052782 | −0.228829 *** |
| | (0.09810) | (0.25266) | (0.01959) |

**Table 10.** *Cont*.

|  | LPAT | TPU | R&D |
|---|---|---|---|
| $R^2$ | 0.937805 | 0.012308 | 0.982656 |
| Adj. $R^2$ | 0.937493 | 0.007353 | 0.982569 |
| F-Statistic | 3005.631 | 2.484015 | 11293.83 |

**Note:** The table shows the estimation of PVAR among patents, TPU, and R&D for the respective developing countries. The data spam is from 1980 to 2020. Standard errors in parentheses. *** $p < 0.01$, ** $p < 0.05$.

### 5.7.3. Impulse Response Function (IRF) Estimates

The impulse response function may analyze the current and future implications of a one-standard-deviation shock on each model variable and the interaction between variables. We analyze the impact of trade policy uncertainty on innovation by using the impulse response function, as presented in Figure 2. We define the policy shock as one standard deviation of escalating trade policy uncertainty over 10 years. This figure shows how financial inclusion and inflation respond to shocks in industrialized nations' endogenous variables. TPU and R&D shocks produce inventive fluctuations. The TPU shock reduces financial inclusion over time. R&D's positive response to the innovation shock confirms bidirectional causality (Barrero et al. 2020).

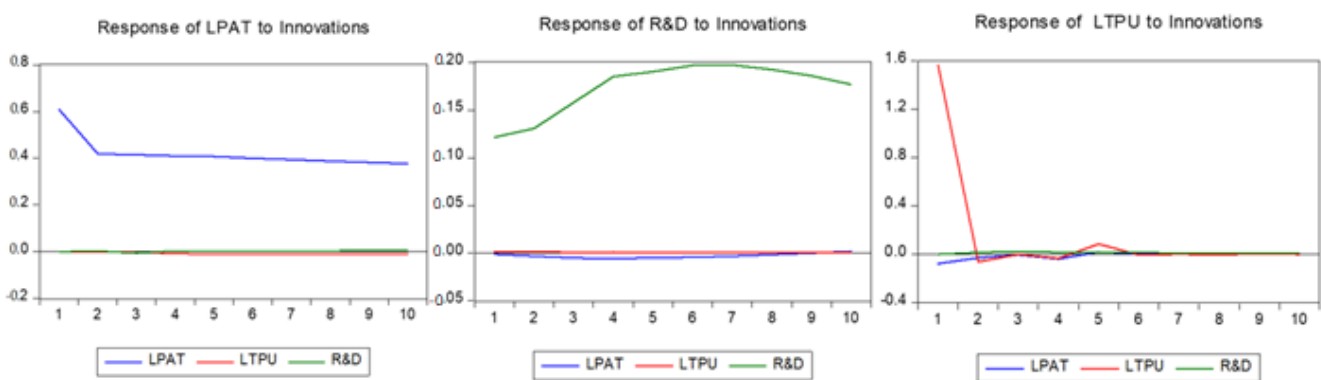

**Figure 2.** Response to Cholesky one S.D (def. adjusted) innovation.

## 6. Conclusions, Policy Implications and Limitations

### 6.1. Conclusions

This study examines the impact of trade policy uncertainty on innovation and R&D spending in developing nations between 1980 and 2020. It employs exogenous and heterogeneous exposure to trade policy uncertainty and comprehensive data on innovation from all medical sectors and developing nations. The PNTR did not increase tariff levels, but innovation was stimulated by reducing tariff uncertainty and securing MFN tariffs through a genuine trade agreement. Reducing tariff uncertainty significantly impacts medical innovation, both economically and statistically, and this effect is indicative of genuine innovation, not merely an increase in patent applications. Additional study into the theoretical framework's mechanisms reveals that the negative innovation response is driven by rising countries, whereas the positive innovation response is driven by R&D spending.

The findings are robust to policy changes and foreign technology inflows in emerging countries. These results highlight trade agreements' vital role in reducing tariff uncertainty and generating economic growth. They are relevant in light of recent events like the US–China trade war, Brexit, and the renegotiation of major trade agreements like NAFTA, which have made tariff uncertainty an essential source for businesses.

Moreover, this study examines the impact of trade policy uncertainty on innovation by using exogenous exposure and controlling for confounding variables. Determine the influence of TPU and R&D expenditures on innovation in developing countries over the

short- and long-term. Long-term innovation fluctuations are statistically significant, both positively and adversely, indicating a rise in medical innovation in developing nations. Results are consistent with the medical innovation hypothesis. The PVAR model between innovation, TPU, and R&D expenses exhibited reverse causality. Innovation and research and development expenditures assist developing nations. TPU generates uncertainty, which reduces the prior year's value.

### 6.2. Policy Implication

Economists have underlined the significance of facilitating medical innovation. Through the elimination of policy uncertainty and, subsequently, the encouragement of innovation in developing and low-income nations, our article suggests that trade liberalization may contribute to economic growth. Understanding the impact of policy uncertainty is crucial for economists and policymakers assessing the efficacy of economic programs. For instance, the time following the 2008 global financial crisis witnessed an increase in trade protectionism. Numerous nations employ non-tariff measures, such as anti-dumping probes, or identify others as "currency manipulators." Recent events, such as the Brexit vote and open calls for protectionist measures by the U.S. government, have all suggested that the future of the global trading system is increasingly questionable. Such protectionist policies may not only impose more significant trade costs but also impede the innovation of businesses, as they generate market uncertainty.

### 6.3. Limitations and Future Study Direction

Concerning the limitations, this study analyzed the developments in the medical industry in 65 developing nations from 1980 to 2020. Consequently, this study's scope and sample size are restricted to medical breakthroughs. The authors recommend that future researchers conduct relevant studies on the innovation of numerous sectors over an updated period or investigate the cross-sectional impact of innovation in developed countries (e.g., the U.S.) by focusing on the same sector.

**Author Contributions:** M.N.S.: Conceptualization, Written the original draft, Data Collection, Data analysis, Design the methodology, and revision. W.L.: Supervision, Conceptualization and Editing. M.U.A.: Data analysis, Design the methodology, Editing and Revision. All authors have read and agreed to the published version of the manuscript.

**Funding:** This research received no external funding.

**Institutional Review Board Statement:** Not applicable.

**Informed Consent Statement:** Not applicable.

**Data Availability Statement:** The datasets used and/or analyzed during the current study have been taken from the database (WIPO, WDI and TRAINS), and made available from the corresponding author on reasonable request.

**Conflicts of Interest:** The authors declare no conflict of interest.

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
