# Peer review of "Trade Policy Uncertainty and Medical Innovation: Evidence from Developing Nations"

_economies, doi:10.3390/economies10090224_

Round 1

Reviewer 1 Report

1. Using 65 developing country data for 1990-2020 the author showed tha6 a reduction in trade policy uncertainty lead to an improvement in medical innovation in a panel DID framework.

2. The topic is interesting but few case studies from countries and sectors would be useful for understanding the essence of this issue more clearly. Only panel study may conceal the stories.

3. It is not clear which method is your chosen method for the paper and which methods are being used for robustness check.

4. The paper has lack of focus. Language needs serious improvement.

5. The abstract is too general. It is hard to understand the research gap and unique contribution of the paper from the abstract as well as from the main body.

6. Sun et al. (2022) is not cited. Reference need to be revised with more complete list.

7. Why did you use ARDL, IV, VAR everything in one place? Which one did you select for your paper?

Author Response

Reviewer 1

  1. Using 65 developing country data for 1990-2020 the author showed that a reduction in trade policy uncertainty lead to an improvement in medical innovation in a panel DID framework.

Answer: Detailed discussed on Page 20, before table 5.

  1. The topic is interesting but few case studies from countries and sectors would be useful for understanding the essence of this issue more clearly. Only panel study may conceal the stories.

Answer: Revised and answered in introduction section on page 5.

  1. It is not clear which method is your chosen method for the paper and which methods are being used for the robustness check.

Answer: Answered on Page 20, second paragraph.

  1. The paper has lack of focus. Language needs serious improvement.

Answer: Language services has been taken from professional and incorporated into the whole document.

  1. The abstract is too general. It is hard to understand the research gap and unique contribution of the paper from the abstract as well as from the main body.

Answer: Abstract has been revised on page 1.

  1. Why did you use ARDL, IV, VAR everything in one place? Which one did you select for your paper?

Answer: Responded on page 17, the last paragraph.

Reviewer 2 Report

The correlation between rising innovation and falling trade policy uncertainty is interesting. But it is not clear that latter causes the former, although I appreciate that the authors were careful to delineate correlation and not argue for causation per se. Secondarily, it is not clear that the findings hold with all trade partners because the US is a very desirable export market. Secondarily, it is one of the world's largest markets for medical products and services. So it stands to reason that reducing tariff uncertainty would increase investment in products that could be sold in the US. So measuring the rise in innovation post-PNTR tells us more about the expectations of investors for selling into the US than it does about the effect of tariff uncertainty on medical innovation in a general sense. That said, we might expect the effect to at least be similar with other large export markets. 

on lines 115-116, be sure to correct the definition of PNTR: it is Permanent Normal Trade Relations

Author Response

Reviewer 2

  1. The correlation between rising innovation and falling trade policy uncertainty is interesting. But it is not clear that latter causes the former, although I appreciate that the authors were careful to delineate correlation and not argue for causation per se.

Answer: Detailed discussed on Page 20, First paragraph.

  1. Secondarily, it is not clear that the findings hold with all trade partners because the US is a very desirable export market.

Answer: Incorporated under Limitations and future study direction of study on page 31 and 32.

As this study is specifically on the developing nations, and also developing nations as their trade partners, either rich or poor. US is excluded from these 65 developing countries, and this is the limitation of my study, it would be discussed in future studies.

  1. Moreover, it is one of the world's largest markets for medical products and services. So it stands to reason that reducing tariff uncertainty would increase investment in products that could be sold in the US. So measuring the rise in innovation post-PNTR tells us more about the expectations of investors for selling into the US than it does about the effect of tariff uncertainty on medical innovation in a general sense. That said, we might expect the effect to at least be similar with other large export markets. 

Answer: Detailed discussed on Page 22, before table 6.

  1. On lines 115-116, be sure to correct the definition of PNTR: it is Permanent Normal Trade Relations.

Answer: Corrected in mentioned lines on page 5.

Round 2

Reviewer 1 Report

Good.